# Characterization of Culturable Mycobiome of Newly Excavated Ancient Wooden Vessels from the Archeological Site of Viminacium, Serbia

**DOI:** 10.3390/jof10050343

**Published:** 2024-05-09

**Authors:** Ivana Djokić, Aleksandar Knežević, Željko Savković, Milica Ljaljević Grbić, Ivica Dimkić, Danka Bukvički, Dragana Gavrilović, Nikola Unković

**Affiliations:** 1Faculty of Biology, University of Belgrade, Studentski Trg 16, 11000 Belgrade, Serbia; b3027_2020@stud.bio.bg.ac.rs (I.D.); knezevica@bio.bg.ac.rs (A.K.); zsavkovic@bio.bg.ac.rs (Ž.S.); jmilica@bio.bg.ac.rs (M.L.G.); ivicad@bio.bg.ac.rs (I.D.); unkovicn@bio.bg.ac.rs (N.U.); 2Institute of Archeology, Kneza Mihaila 35/IV, 11000 Belgrade, Serbia; gavrilovicdragana@yahoo.com

**Keywords:** archeological wood, *Ascomycota*, biodegradative plate assays, buried monoxyle, buried shipwreck, conservation, enzymes, Mammoth Park, *Penicillium*, soft-rot fungi

## Abstract

Two ancient wooden vessels, specifically a monoxyle (1st century BCE to 1st century CE) and shipwreck (15th to 17th century CE), were excavated in a well-preserved state east of the confluence of the old Mlava and the Danube rivers (Serbia). The vessels were found in the ground that used to be river sediment and were temporarily stored within the semi-underground exhibition space of Mammoth Park. As part of the pre-conservation investigations, the primary aim of the research presented was to characterize the culturable mycobiomes of two excavated wooden artifacts so that appropriate conservation procedures for alleviating post-excavation fungal infestation could be formulated. Utilizing culture-based methods, a total of 32 fungi from 15 genera were identified, mainly *Ascomycota* and to a lesser extent *Mucoromycota sensu stricto*. Soft-rot *Ascomycota* of genus *Penicillium*, followed by *Aspergillus* and *Cephalotrichum* species, were the most diverse of the isolated fungi. Out of a total of 38 isolates, screened on 7 biodegradation plate assays, 32 (84.21%) demonstrated at least one degradative property. *Penicillium solitum* had the highest deterioration potential, with a positive reaction in 5 separate plate assays. The obtained results further broaden the limited knowledge on the peculiarities of post-excavation soft-rot decay of archaeological wood and indicate the biochemical mechanisms at the root of post-excavation fungal deterioration.

## 1. Introduction

Humanity has been using wood for various purposes since ancient times. Due to its mechanical durability, alongside stone, wood was one of the earliest materials used and is still in use today for the construction of infrastructure facilities, support columns, houses, wooden vessels, art objects, etc. However, archeological wooden objects are rarely found, since the organic chemical nature of wood limits its longevity, unless protected by an environment that limits microbial activity [1]. Since archaeological wood is usually already structurally damaged, special attention is required when it comes to its handling, i.e., study and conservation. Therefore, it is necessary to improve the understanding of the (bio)chemical processes and changes that affect archaeological wooden artifacts during aging and degradation in order to develop effective treatments that can be applied for their preservation [2,3].

Wood is an easily processed and shaped material that is suitable for use in construction and art due to its structure, strength, and plasticity; however, the biodeterioration of wooden objects is a serious problem when it comes to durability [1]. Since all objects of with cultural heritage made of wood undergo constant chemical transformations, they deteriorate over time, either slowly or progressively. The degree of deterioration depends on exposure to various biotic and abiotic factors. Even though abiotic factors can potentially cause wood to deteriorate, they are thought to have a negligible effect compared to biotic factors [4]. Biotic factors refer to the process of rotting caused by organisms for which wood is a habitat or nutrient source [1,5]. Wood consists of lignin (10–20%), cellulose (30–35%), and hemicellulose (25–30%), three factors which makes up about 70% of the total mass. Additionally, it contains lipids, proteins, water, and other compounds, whose percentages vary between species. These factors are responsible for the resistance and strength of the wood mass [6,7]. However, two dominant polymers and extractives in the cell walls of plants are the main sources of nutrients for numerous organisms, especially fungi, and bacteria, which makes wood highly bioreceptive and biodegradable. The chemical composition of core and bark, which varies with the plant species, age, and conditions prevailing in the habitat, determines how susceptible the wood will be to decay [8]. The deterioration of wooden objects also depends to the greatest extent on environmental conditions, primarily relative humidity and temperature [9].

Wood-degrading microorganisms play a leading role in wood deterioration. Among them, fungi are of the greatest importance since they can colonize different substrates due to their enormous enzymatic activity and their ability to grow at low values of water content in different materials [10,11]. Hence, fungi are the primary organisms able to break down wood mass or cause esthetic alterations. Based on ecological succession and the type of damage to the material, fungi that inhabit wooden objects can be divided into (1) fungi that cause pigmentation, and (2) fungi that cause rot. Consequently, the main causes of deterioration processes are those fungi that produce enzymes that play a dominant role in wood decomposition [1,12]. Generally, three types of fungi that damage wood can be distinguished: brown rot, white rot, and soft rot. However, in extreme conditions that are specific to archeological wood, which include high moisture content, low temperatures, and/or depleted oxygen, wooden biomass is mostly attacked by soft-rot fungi. These fungi are mostly *Ascomycota* and are present in various terrestrial and aquatic environments. The changes that occur to wooden objects under the action of soft-rot fungi are characterized by cavity formation within the cell wall due to cellulose and hemicellulose degradation, which makes the material brittle in the long term [13].

Although the possibility of finding wooden objects of cultural and historical importance in a solidly preserved state is extremely small, in March 2020, due to a series of significant environmental circumstances, a well-preserved shipwreck (dated between the 15th and the 17th centuries CE) and a monoxyle, which is a simple vessel obtained by hollowing out a single tree trunk (dated between the 1st century BCE and the 1st century CE), were discovered east of the confluence of the old Mlava and the Danube (Serbia) in ground that used to be river sediment [14].

These wooden objects were found between 6 and 10 m in terms of the profile of the coal mine. They were found in anoxic conditions, which made it possible to preserve the composition of the wood mass for so many years. The monoxyle and the shipwreck were lying on the ground that used to be the river sediment. Since there was a former path for the river course, created after the river course changed, this area became a moor. The soil layer above the objects was created by the combination of aeolian and fluvial sediments. All these environmental conditions, particularly depleted oxygen, resulted in extremely slow degradation processes, allowing the monoxyle and shipwreck to be preserved for centuries. After the excavation, due to the COVID-19 pandemic, the objects were temporarily stored within the semi-underground exhibition space of Mammoth Park.

Since waterlogged and buried archaeological woods undergo slow erosion bacteria-induced degradation processes in anoxic conditions, they become prone to microbial attack post-excavation by more aggressive microorganisms from the environment, primarily filamentous fungi [1,15]. To successfully protect rare wooden objects, i.e., artifacts with importance to humankind, it is necessary to know the deterioration capabilities of fungi, which alter the chemical structure of wood by degrading its three main components, therefore making it weaker and more susceptible to further decay. Therefore, the principal aim of this study was to characterize the composition and degradative potential of culturable mycobiomes of investigated wooden artifacts, ensuring that adequate proposals for conservation treatments could be made.

## 2. Materials and Methods

### 2.1. Study Site and Sampling Points

This study was conducted at the Roman archeological site of Viminacium, a cultural property on the UNESCO Tentative list located 3 km south of the Danube within the administrative borders of the City of Požaervac (Serbia) (Figure 1a) [16]. Sampling was performed before any conservation treatments on two wooden artifacts, a monoxyle (Figure 1b,c) and shipwreck (Figure 1d). These were temporarily stored post-excavation within the indoor semi-underground exhibition space of Mammoth Park—an underground chamber that spans 1200 square meters, with a wooden arched-roof construction. Light tunnels illuminate the space with natural daylight. The walls of this space are made of earth, and the floor is covered with sand [16]. 

In situ observations of investigated wooden artifacts, i.e., the monoxyle and shipwreck, made one year after their excavation and placement within the Mammoth Park, showed distinct differences between the two artifacts in terms of the visibility of biofilm-induced deterioration symptoms. Namely, wooden surfaces of monoxyle fragments were covered with clearly visible (1) patches of compact ochre fungal growth and diffusely spread mycelial networks of *Basidiomycota*. They also showed (2) greyish tufts composed of dense and erect reproductive structures of *Mucoromycota*. Sampling for the mycological analysis was carried out at 11 sampling points (M1–M11) on the monoxyle, all characterized by the presence of distinct fungal-induced alterations of the wooden surface, and four sampling points on shipwreck (B1–B4), which lacked any visible symptoms of fungal infestation but were spots characterized by being in a highly dilapidated state.

### 2.2. Indoor Microclimate and Moisture Content Measurements

The temperature (T °C) and relative humidity (RH %) of the Mammoth Park ambient were measured every 30 min during the 18-month period (from 21 June 2021 to 9 November 2022). For this purpose, a Testo 176P1 data logger (Testo SE & Co. KGaA, Titisee-Neustadt, Germany) was installed on the wooden beam at a height of 250 cm from the floor. Furthermore, the moisture content (%) of wooden artifacts was measured using a Testo 606-2 measuring instrument (Testo SE & Co. KGaA, Titisee-Neustadt, Germany), which was set to wood mode.

### 2.3. Isolation and Identification of Culturable Fungi

Sterile swabs were used to collect samples from the monoxyle and shipwreck. In laboratory conditions, samples were spread onto a potato dextrose agar (PDA, TM Media, Rajasthan, India) plates, supplemented with 500 mg L^−1^ Streptomycin sulfate salt (Sigma Aldrich, St. Louis, MO, USA) to prevent bacterial growth. The inoculated media were incubated in a thermostat chamber (UE 500, Memmert, Schwabach, Germany) at 25 ± 2 °C for 7 days. After the incubation period, morphologically different colonies were reinoculated on PDA plates in triplicate and incubated under the same conditions to obtain axenic cultures. The isolated fungi were identified based on the macroscopic features of colonies, observed under a Stemi DV4 stereomicroscope (Carl Zeiss, Oberkochen, Germany), and the micromorphological characteristics of reproductive structures, analyzed with the optical microscope Zeiss Axio Imager M1 (Carl Zeiss, Oberkochen, Germany) using AxioVision Release 4.6 software. For identification, appropriate identification keys were used [17,18,19,20].

The molecular identification of fungal isolates was additionally performed to confirm the identification results based on isolate morphology. DNA isolation was performed according to the instructions of the Quick-DNA Fungal/Bacterial Miniprep Kit (ZYMO RESEARCH USA, Irvine, CA, USA), removing 50–100 mg of marginal mycelia from 7-day-old cultures. The ITS region (ITS1/4) and BenA (BT2a/BT2b) were used as markers. PCR was performed according to the procedure previously described by Savković et al. [21]. The obtained PCR amplicons were separated via electrophoresis on agarose gel, purified with EXTRACT ME DNA CLEAN-UP KIT (BLIRT S.A, Gdańsk, Poland), and sent for sequencing (Eurofins Genomics Europe Sequencing GmbH, Ebersberg, Germany). The obtained sequences were aligned with appropriate sequences from the National Center for Biotechnology Information (NCBI) database using the BLAST program (BLAST+ 2.15.0). Furthermore, all the sequences were aligned using the ClustalW algorithm of MEGA11 software. The resulting tree was constructed using neighbor-joining phylogeny. Repeats of 1000 times were used to obtain branches with the highest confidence possible.

### 2.4. Biodegradation Plate Assays

Fungal isolates obtained from the monoxyle and shipwreck, deposited to the Culture Collection of the University of Belgrade—Faculty of Biology (BEOFB), were tested for the following degradative capabilities: pigment production, acid metabolite production, cellulase production, hemicellulase production, and ligninolytic enzyme production (laccases, Mn-oxidizing peroxidases, and lignin peroxidases). All biodegradative plate assays were performed in triplicate. The obtained results were summarized in the form of five-way and three-way Venn diagrams, constructed using the web-based tool ^InteractiVenn^ [22] (https://www.interactivenn.net/, accessed on 15 December 2023).

To qualitatively determine extracellular pigment production, fungal isolates were tested on minimal Czapek–Dox agar (mCzA) according to the instructions of Borrego et al. [23]. In 1 L of distilled water, we dissolved NaNO_3_, 2 g; K_2_HPO_4_, 1 g; MgSO_4_ × 7H_2_O, 0.5 g, glucose (Sigma-Aldrich, St. Louis, MO, USA), 1 g; and agar (Lab M, Heywood, UK), 20 g. The pH level was adjusted to 5.5. The medium was sterilized at 114 °C for 25 min. Plates were inoculated and incubated at 25 ± 1 °C for 7 days. The coloration of a transparent medium was considered to indicate a positive reaction.

For the detection of acid metabolite production, we used creatine sucrose agar (CREA), prepared according to the instructions of Samson et al. [19]. The medium contains the pH indicator Bromocresol purple (5′,5″-dibromo-o-cresolsulfophthalein, Centrohem Ltd., Stara Pazova, Serbia), which changes color to yellow at pH values below 5.2. It was prepared with 1 L of distilled water, in which we dissolved KCl, 0.5 g; MgSO_4_ × 7H_2_O, 0.5 g; FeSO_4_ × 7H_2_O, 0.01 g; K_2_HPO_4_ × 3H_2_O, 1.3 g; creatinine (Sigma-Aldrich), 3 g; sucrose (Merck, Darmstadt, Germany), 30 g; agar, 15 g; and Bromocresol purple, 0.05 g. pH values were between 7.8 and 8.2. Inoculated plates were incubated at 25 ± 1 °C for 7 days. A visible change in the color of the media around the growing colonies, from purple to yellow, was considered to indicate a positive reaction.

Carboxymethyl cellulose sodium salt (CMC, Sigma-Aldrich) agar plates were used for cellulase detection in selected isolates according to the recommendations of Liang et al. [24]. In 1 L of distilled water, the following substances were dissolved: (NH_4_)_2_SO_4_, 1.4 g; KH_2_PO_4_, 2 g; CaCl_2_, 0.3 g; MgSO_4_ × 7H_2_O, 0.3 g; FeSO_4_ × 7H_2_O, 0.005 g; CoCl_2_, 0.002 g; MnSO_4_, 0.001 g; ZnSO_4_, 0.0014 g; urea (Lach-Ner, Zagreb, Croatia), 0.3 g; yeast extract (Biolife, Milan, Italy), 0.25 g; peptone (Lab M), 0.75 g; CMC, 10 g; and agar, 20 g. pH values were adjusted to 5.0. The inoculated plates were incubated at 25 °C ± 1 for 7 days. After the incubation period, plates were stained with 0.1% Congo red (TCI Ltd., Gurgaon, India) solution (10 min), after which they were de-stained with 1 M NaCl to observe the decolorized zones around colonies of the isolates that produced cellulolytic enzymes. The presence of the transparent zone was considered to be a positive result. 

The screening of hemicellulase-producing fungi was carried out on Xylan agar plates that were prepared as CMC agar plates, with Xylan from beach wood (Roth, Karlsruhe, Germany) used instead of CMC. The plates were incubated at 25 °C ± 1 for 7 days. After the incubation period, all plates were stained with 0.1% Congo red solution for 10 min and washed with 1 M NaCl. The presence of the transparent zone around the colony was considered to indicate a positive result [24].

Three different media were used to assess the ability of the isolates to degrade lignin: we used a medium containing 2-2′-Azino-bis-[3-ethyl benzthiazoline-6-sulfonic acid] (ABTS, TCI Ltd.) for laccase detection, a medium containing Azure II (Sigma-Aldrich, St. Louis, MO, USA) for lignin peroxidize (LiP) detection, and a medium containing Phenol red for Mn-oxidizing peroxidase (MnP) detection.

The medium used for laccase activity detection contained, per 1 L of distilled water, the following elements: ABTS, 1.644 g; PDA, 39 g; and CuSO_4_, 0.160 g. The pH was adjusted to 4.5. The medium was sterilized at 114 °C for 25 min. The inoculated plates were incubated at 25 ± 1 °C for 7 days. The formation of a blue-green zone in the medium around the colony was considered to indicate a positive result [25].

The medium used for MnP detection contained Phenol-red (Sigma-Aldrich) as an indicator of enzyme presence. In 1 L of distilled water, we dissolved the following after sterilization: NaNO_3_, 3 g; K_2_HPO_4_, 1 g; MgSO_4_ × 7H_2_O, 0.5 g; KCl, 0.5 g; FeSO_4_ × 7H_2_O, 0.01 g; sucrose, 30 g; agar, 15 g; Penol red, 0.0125; and H_2_O_2_, 1 mL of 40 mM. The formation of a yellow zone in the medium around the fungal colony was considered to be a positive result [26].

Working according to Sharma et al. [27], the medium used for LiP detection contained Azure II, which was decolorized in the presence of LiP enzymes. In 1 L of distilled water, the following substances were dissolved: KH_2_PO_4_, 1 g; NH_4_NO_3_, 0.5 g; MgSO_4_ × 7H_2_O, 0.5 g; FeSO_4_ × 7H_2_O, 0.01 g; MnSO_4_ × H_2_O, 0.01 g; CaCl_2_ × 7H_2_O, 0.01 g; CuSO_4_ × 5H_2_O, 0.01 g; yeast extract, 0.1 g; agar, 20 g; and Azure II, 0.1 g. Sterilization was performed at 114 °C for 25 min. The inoculated plates were incubated at 25 ± 1 °C for 7 days. The occurrence of discoloration in the medium around the fungal colony was considered to indicate a positive result. 

## 3. Results

### 3.1. Microclimate of the Mammoth Park Ambient

A total of 24,243 measurements of the temperature and relative humidity of the ambient of Mammoth Park were conducted during the 18-month period. The temperature ranged from 6.0 °C to 21.0 °C, while the relative humidity was in a range from 83.9% to 99.9% (Figure 2). The mean values of temperature and relative humidity were calculated to be 14.6 °C and 98.4%, respectively. 

Moisture content values of wooden artifacts were in the range from 13.4 to 46.6% and 14.4 to 23.3% for the monoxyle and shipwreck, respectively. The measured value on sampling point M7 on the monoxyle was out of the measuring scope of the instrument.

### 3.2. Diversity of Cultivable Mycobiomes

The total culturable mycobiome, isolated from the surfaces of the monoxyle and shipwreck, was composed of 32 taxa belonging to 15 genera (38 isolates in total) (Table 1). Higher diversity was documented on the monoxyle, with 25 isolated species, while 13 species were identified from shipwreck samples. *Absidia repens*, *Lunasporangiospora selenospora*, *Mortierella alpina*, and *Mucor aligarensis* are representatives of the phylum *Mucoromycota*, while the other isolated fungi are members of the dominant phylum *Asomycota*. Of the 12 identified species, the most diverse were fungi of the genus *Penicillium*, followed by species of genera *Aspergillus* and *Cephalotrichum*, with four species each. All other genera were represented with only one species.

The neighbor-joining tree of the ITS region is shown in Figure 3 and the *BenA* tree is shown in Figure 4. All *Aspergilli* and *Penicillia* clustered together in a well-supported *Eurotiales* clade (bootstrap value (bs) = 100). *Beauveria fellina*, *Trichoderma zeloharzianum*, *Sarocladium spinificis*, *Fusarium langsethiae*, and *Lecanicilium dimorphum* clustered together in a *Hypocreales* clade (bs value = 59). Likewise, *Cephalotrichum* and *Apiospora* isolates clustered together with *Hypocreales* isolates in the *Sordariomycetes* clade (bs value = 98). All *Ascomycota* clustered together (bs value = 97), as did the isolates of the *Mucorales* and *Mortierellales* orders in the *Mucoromycota* clade (bs value = 76).

### 3.3. Biodegradative Profile of Isolated Fungi

Out of a total of 38 isolates screened using 7 plate assays, 32 (84.21%) demonstrated at least one degradative property, while 6 (15.79%) isolates lacked any activity (Table 1 and Figure 5). The majority of isolates demonstrated laccase and hemicellulolytic activity, shown by 21 (65.63%) and 20 (62.5%), respectively. The somewhat smaller number of isolates demonstrated acid metabolite production (14; 43.75%), cellulolytic (11; 34.38%), and Mn-oxidizing peroxidase (10; 31.25%) activities. Only 4 isolates showed pigment production (12.5%), while none of the tested isolates from wooden artifacts showed lignin peroxidase activity. *Penicillium solitum* (BEOFB0110902) showed the highest deterioration potential, with a positive reaction in 5 out of 7 plate assays (MnP, Lac, HA, AP, CA), followed by *P. aurantiogriseum*, *P. flavigenum*, *P. griseofulvum*, *P. speluncae*, *Cephalotrichum asperulum*, and *C. dendrocephalum*, with four documented activities.

## 4. Discussion

Nowadays, there is a lack of data regarding the origin and deteriogenic role of the fungal communities infesting archaeological wooden artifacts in the post-excavation period, resulting in difficulties in foreseeing the course of the deterioration process and consequently the application of appropriate conservation treatments. Research on non-archaeological wood indicates that the wood-inhabiting *Ascomycota* and *Basidiomycota* isolates are the primary agents of deterioration; however, due to the numerous peculiarities of archaeological wooden artifacts after their modification during long-term exposure to anoxic sediments and the specific environments in which these objects are kept post-excavation, comprehensive knowledge on fungal-induced decay in terrestrial environment is not entirely attainable [28]. The results from culturable mycobiomes of the investigated wooden artifacts, i.e., a monoxyle and shipwreck, dominated by soil-borne *Ascomycota,* were in complete accordance with previous studies, in which fungi isolated from various archeological wood samples were related to the aerobic, post-excavation periods, rather than the anoxic sediment environment from which artifacts were excavated [29,30]. Moreover, soil tends to be the main source of fungal colonizers for wooden objects in contact with the ground [31]. The composition of culturable mycobiomes of the investigated artifacts is a result of the distinct semi-hypogean environment of Mammoth Park, i.e., a sand base and walls of excavated soil, accounting for the high diversity of *Mucoromycota* (*Absidia repens*, *Lunasporangiospora selenospora*, *Mortierella alpina,* and *Mucor aligarensis*) and *Ascomycota* of genera *Penicillium*, *Aspergillus,* and *Cephalotrichum*. The presence of 12 *Penicillium* species, especially *P. chrysogenum*, is indicative of their potential role in the deterioration of studied wooden artifacts since fungi of this genus are reported to be soft-rot fungi, which are already associated with the decay of archeological wood [32,33]. Furthermore, *Aspergillus*, *Fusarium*, and *Trichoderma* are also reported to be soft-rot deteriogens of archaeological wood from the Middle Cemetery at Abydos, Egypt [30]. The soft-rot fungi are mostly able to colonize wooden substrates in conditions where the growth of white-rot and brown-rot fungi is inhibited by high moisture content, low aeration, and low or high temperatures. High moisture content, up to 45.6%, documented for the monoxyle far exceeds the value of 20% considered optimal for the suppression of microbial infestation [1]. 

On the other hand, moisture content for the shipwreck was in the recommended range, which corresponded to the dry appearance of the artifact and the absence of visible fungal growth. The role of the microbial community in the decay of archeological wood was also dependent on the environmental conditions, with archeological wooden artifacts being especially susceptible to biodeterioration after excavation if not handled and restored properly [34]. Hence, to avoid post-excavation fungal infestation, special attention should be taken during the conservation process.

Based on the in vitro plate assays, it can be stated that the majority of isolated fungi possessed the potential to esthetically and structurally impair the investigated wooden artifacts in observed environmental conditions. Pigment production, as one of the primary mechanisms of irreversible discoloration and artifact staining, was observed in cultures of *Penicillium*, *Cephalotrichum*, and *Alternaria* species via the coloration of the used transparent nutrient medium. *Penicillium* species are among the most prominent pigment producers, with more than 73% of *Penicillium* isolates performing this role. Among them, *P. glabrum*, obtained from the historic buildings of Havana, demonstrates the capacity to produce pigments on wooden artifacts [35]. Interestingly, none of the fungi that demonstrate pigment production in vitro can be associated with the distinctive pink discolorations observed on the surface of the monoxyle. On the other hand, the capability of fungi to produce inorganic (carbonic) and organic acids (gluconic, citric, oxalic, fumaric, etc.) and indirectly induce wood degradation via the enhancement of fungal lignocellulolytic enzymes is one of the crucial mechanism responsible for the alterations of both non- and the archeological wood [21,36]. The acidification of the wood via organic acids reduces the viscosity of the hemicellulose and cellulose, and is subsequently followed by the secretion of hydrolytic enzymes that decompose wood components. The obtained results emphasize that fungi of the *Penicillium* genus are the most prominent acid producers, with 10 out of 15 tested isolates changing the color of Bromocresol from purple to yellow. Among fungi of this genus, *P. aurantiogriseum*, *P. chrysogenum*, *P. griseofulvum*, and *P. solitum* stood out as the strongest producers. An overview of the literature pointed to numerous acids produced by these species: terrestric and penicillic acids (*P. aurantiogriseum*); secalonic, emocid, α-aminoadipic, and shikimic acids (*P. chrysogenum*); and α-cyclopiazonic and dehydrofulvic acids (*P. griseofulvum*) [37,38,39]. 

In the post-excavation period, when archaeological wooden artifacts are exposed to a significantly altered environment, the deterioration process is accelerated via the carbohydrate metabolism of a secondary microbial community, which attacks both cellulosic and lignin components [40]. Fungal-produced extracellular hydrolytic enzymes, i.e., cellulases, hemicellulases, and ligninolytic enzymes (Mn-oxidizing peroxidases, lacasse, lignin peroxidase, etc.), are primary agents responsible for breaking down the main components of wood into simple molecules, used as nutrient sources ([bio]chemical assimilatory biodeterioration). Many soft-rot fungi, such as representatives of *Trichoderma*, *Fusarium*, and *Penicillium* genera, possess a complete set of enzymes, capable of the efficient degradation of cellulose [41]. Isolates of all three genera were screened for cellulase production; however, only four *Penicillium* species (*P. flavigenum*, *P. griseofulvum*, *P. solitum*, and *P. speluncae*) demonstrated various levels of cellulolytic production, while cultures of *T. zeloharzianum* and *F. langsethiae* lacked observable transparent zones around colonies grown on CMC agar plates. Consistent results, with high levels of cellulase production by *P. solitum* and moderate levels by *P. flavigenum* and *P. griseofulvum*, were reported by Yoon et al. [42]. In a study by Zyani et al. [41], high levels of cellulase production were documented for three *Penicillium* species (*P. chrysogenum*, *P. commune*, and *P. granulatum*), while weak production was observed for *P. crustosum*, and *P. expansum*. These species were isolated from 450-year-old wooden constructions at the Medina of Fez (Morocco) and screened on CMC agar plates. Furthermore, cellulase production in isolates from monoxyle and shipwreck was also documented for *Beauveria felina*, *Sarocladium spinificis*, *Cephalotrichum asperulum, C. cylindricum*, *Lunasporangiospora selenospora,* and *Mortierella alpina*. These are all genera whose members, according to the literature review, possess variable levels of cellulase production [43,44,45,46].

Although members of the *Basidiomycota* and *Ascomycota* phyla, especially those inhabiting soil and wood, are primarily known for their capacity to produce ligninolytic enzymes, it has been demonstrated that representatives of the phylum *Mucoromycota* can also possess this capability [47,48,49]. The obtained results suggest that fungal communities on investigated objects, reflected in culturable mycobiota, are mostly characterized by taxa capable of producing ligninolytic enzymes. This was confirmed vuia biodegradative assays, in which 23 out of 38 isolates were capable of synthesizing one or two ligninolytic enzymes. This activity was not demonstrated at all for fungi of genera *Mucor*, *Scytalidium,* and *Trichoderma*, although the literature data suggest that species from these genera are capable of ligninolytic enzyme production [50,51,52]. Furthermore, *Lunasporangiospora selenospora* (formerly *Mortierella selenospora*) showed no ligninolytic activity, even though it was previously demonstrated that fungi of this genus can be good laccase producers [49]. Additionally, whole-genome sequencing of *M. elongata* demonstrated the presence of laccase genes [48]. However, to the best of our knowledge, this study is the first report of laccase activity in *M. alpina*. Moreover, the absence of ligninolytic activity in *Beauveria felina* is in accordance with results obtained by Kameshwar and Qin [48]. Finally, fungi capable of producing both ligninolytic and cellulolytic enzymes are a more serious threat to vulnerable wooden artifacts since these enzymes can act synergistically in the process of wood decomposition [53,54]. A total of 16 out of 38 isolates, obtained from the monoxyle and shipwreck, showed combined ligninolytic and cellulolytic/hemicellulolytic activity, including species from genera *Aspergillus*, *Cephalotrichum*, *Lecanicillium*, *Mortierella*, *Penicillium* and *Sarocladium*. Among them, *Cephalotrichum* species can be considered as potentially the highest threat since three out of four isolated species showed this combined activity. 

## 5. Conclusions

Insight into the culturable mycobiome and its biodegradative potential, studied in vitro on a set of plate assays, was obtained as part of a pre-conservation investigation, with results used as indicators in the selection of the most appropriate conservation treatments for alleviating post-excavation fungal infestation. The results point out that almost all the fungi documented as part of the culturable mycobiome (32 out of a total of 38; 84.21%) are able to produce one or several deterogenic metabolites when cultured on appropriate nutrient media in vitro. If the documented fungi develop on the studied wooden artifacts, there is a possibility that harmful metabolites will be produced, resulting in further structural and aesthetical impairments. However, metabolite production is not universal but dependent on other factors, such as the nature of the substrate on which fungi grow. As a result of this, metabolite production may not even occur. Bearing all this in mind, it is necessary to closely monitor the condition of the artifacts post-excavation and control environmental parameters they are exposed to in order to prevent the development of fungi and the possible production of deteriogenic metabolites.

Conservation and restoration research on the fragile artifacts is still ongoing, with the exhibits now in a stable condition and lacking any visible microbial growth. Furthermore, the presented results emphasize the need for additional research on the diversity and deterioration capacity of the post-excavation terrestrial soft-rot fungi infesting vulnerable archeological wooden artifacts, since the preservation of such wood against biodegradation is different than that of regular wood. In this way, the threat thriving mycobiome potentially poses to the long-term preservation of delicate artifacts will be better understood and the appropriate conservation procedure for alleviating post-excavation fungal infestation can be successfully implemented. 

## Figures and Tables

**Figure 1 jof-10-00343-f001:**
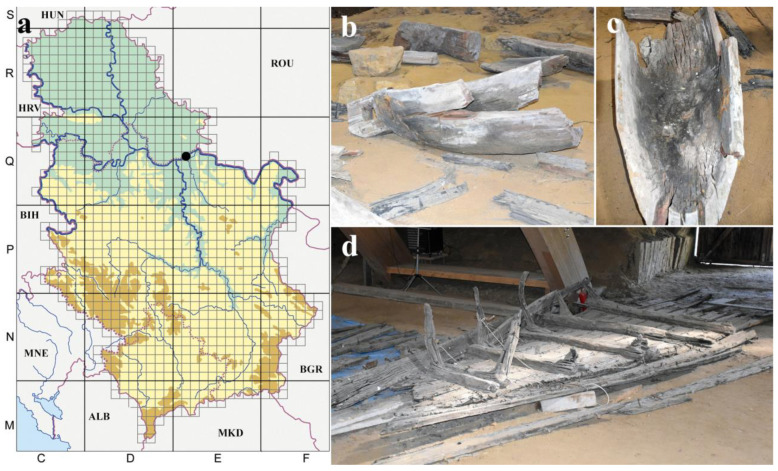
Archeological wooden artifacts in the Mammoth Park of the archaeological site “Viminacium” before conservation treatment: (**a**) UTM map of Serbia with marked location of the archaeological site “Viminacium (black dot)”; (**b**,**c**) monoxyle; (**d**) shipwreck. Country abbreviations: HUN—Hungary; ROU—Romania; BGR—Bulgaria; MKD—North Macedonia; ALB—Albania; MNE—Montenegro; BIH—Bosnia and Herzegovina; HRV—Croatia.

**Figure 2 jof-10-00343-f002:**
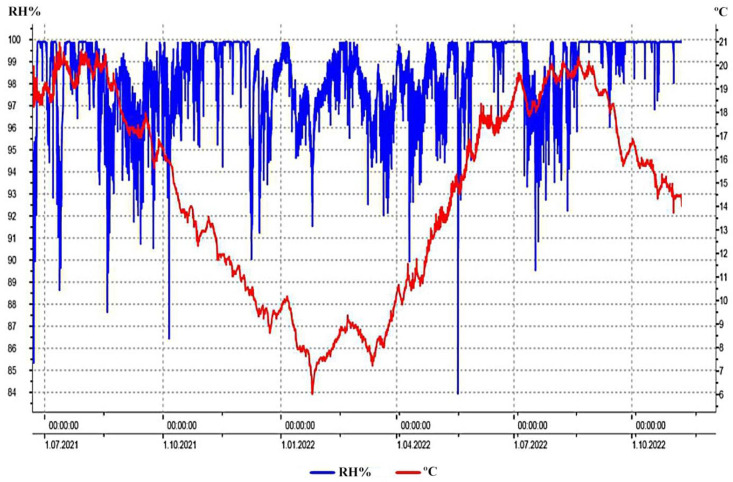
Fluctuations of temperature (T °C; red line) and relative air humidity (RH%; blue line) of the Mammoth Park ambient within the 18-month measuring period.

**Figure 3 jof-10-00343-f003:**
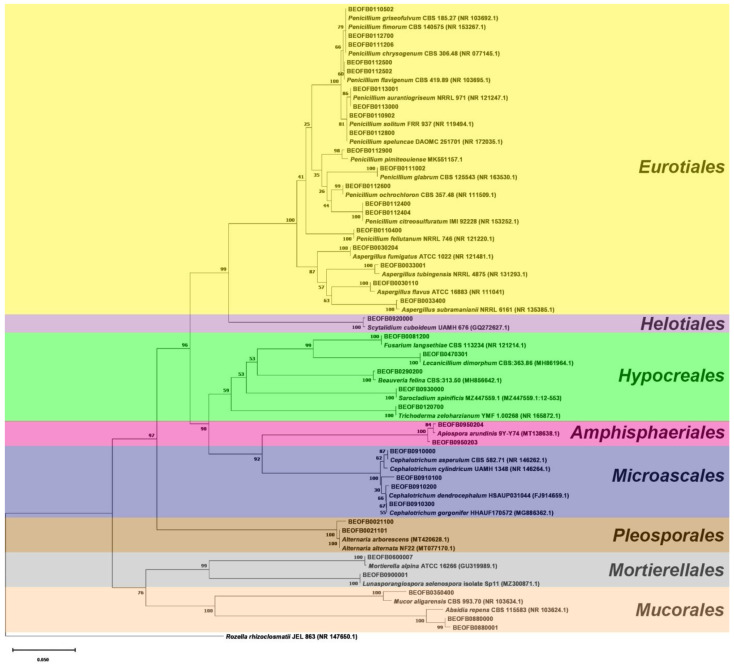
The ITS region neighbor-joining tree of fungi isolated from the monoxyle and shipwreck; *Rozella rhizoclosmatii* JEL 863 (NR 147650.1) was used as the outgroup.

**Figure 4 jof-10-00343-f004:**
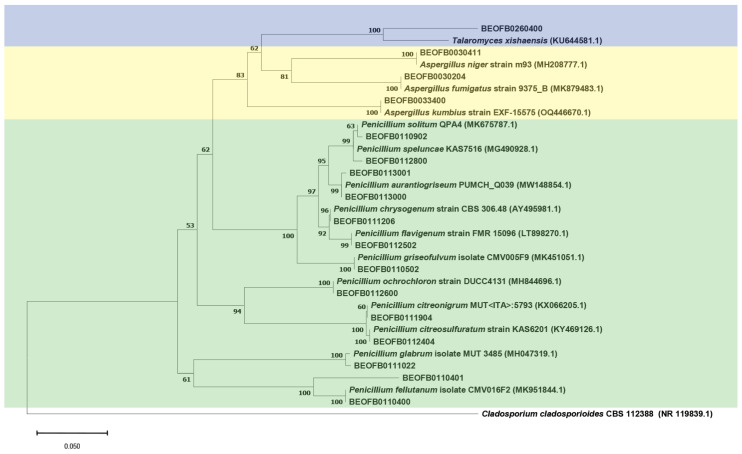
The *BenA* region Neighbor-joining tree of fungi isolated from the monoxyle and shipwreck; *Cladosporium cladosporioides* CBS 112348 (NR 119839.1) was used as the outgroup.

**Figure 5 jof-10-00343-f005:**
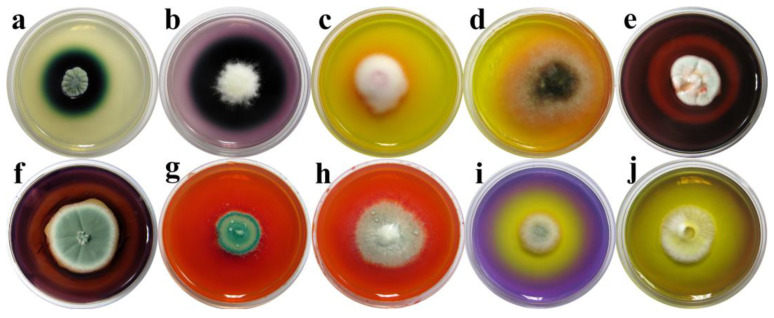
Biodegradative profile of screened fungal isolates. Activity: (**a**,**b**) laccase activity; (**c**,**d**); Mn-oxidizing peroxidase activity; (**e**,**f**) hemicellulolitic activity; (**g**,**h**) cellulolitic activity; and (**i**,**j**) acid metabolite production. Fungi: (**a**,**g**,**i**). *Penicillium solitum*; (**b**). *Apiospora arundinis*; (**c**). *Lecanicillium dimorphum*; (**d**). *Alternaria* sp.; (**e**). *Penicillium citreosulfuratum*; (**f**,**h**). *Penicillium glabrum*; (**j**). *Penicillium pimiteouiense*.

**Table 1 jof-10-00343-t001:** Diversity and biodegradative profile of culturable fungi isolated from monoxyle (M) and shipwreck (B).

Number	Strain	Sample Site	Isolate	Biodegradative Profile
1	BEOFB0880000	M1	*Absidia repens*	HA
2	BEOFB0880001	B1	*Absidia repens*	MnP, Lac, AP
3	BEOFB0021100	M10	*Alternaria* sp.	MnP, Lac
4	BEOFB0021101	B4	*Alternaria* sp.	HA, AP, PP
5	BEOFB0950204	M7	*Apiospora arundinis*	Lac
6	BEOFB0950203	B3	*Apiospora arundinis*	Lac
7	BEOFB0030110	M11	*Aspergillus flavus* var. *flavus*	Lac, HA
8	BEOFB0030204	B4	*Aspergillus fumigatus*	-
9	BEOFB0033400	M9	*Aspergillus kumbius*	-
10	BEOFB0030411	M11	*Aspergillus niger*	AP
11	BEOFB0290200	M10	*Beauveria felina*	CA
12	BEOFB0910000	M3	*Cephalotrichum asperulum*	MnP, Lac, HA, CA
13	BEOFB0910100	M6	*Cephalotrichum cylindricum*	Lac, HA, CA
14	BEOFB0910200	B1	*Cephalotrichum dendrocephalum*	MnP, HA, AP, PP
15	BEOFB0910300	M9	*Cephalotrichum gorgonifer*	HA
16	BEOFB0081200	B4	*Fusarium langsethiae*	Lac
17	BEOFB0470301	M2	*Lecanicillium dimorphum*	HA, CA, MnP
18	BEOFB0900001	M3	*Lunasporangiospora selenospora*	HA, CA
19	BEOFB0600007	M2	*Mortierella alpina*	MnP, Lac, CA
20	BEOFB0350400	B2	*Mucor aligarensis*	-
21	BEOFB0113000	M10	*Penicillium aurantiogriseum*	Lac, HA, AP
22	BEOFB0113001	B3	*Penicillium aurantiogriseum*	Lac, HA, AP, PP
23	BEOFB0111206	B1	*Penicillium chrysogenum*	HA, AP
24	BEOFB0111904	M2	*Penicillium chrysogenum*	Lac, HA
25	BEOFB0112404	B1	*Penicillium citreosulfuratum*	Lac, HA
26	BEOFB0110400	M1	*Penicillium fellutanum*	Lac
27	BEOFB0110401	M6	*Penicillium fellutanum*	Lac, HA
28	BEOFB0112500	M3	*Penicillium flavigenum*	MnP, Lac, HA, AP
29	BEOFB0112502	B3	*Penicillium flavigenum*	AP, HA, CA
30	BEOFB0111002	B4	*Penicillium glabrum*	AP, HA, PP
31	BEOFB0110502	M7	*Penicillium griseofulvum*	AP, CA, MnP, Lac
32	BEOFB0112600	M3	*Penicillium ochrochloron*	-
33	BEOFB0112900	M10	*Penicillium pimiteouiense*	Lac, AP
34	BEOFB0110902	B3	*Penicillium solitum*	MnP, Lac, HA, AP, CA
35	BEOFB0112800	M6	*Penicillium speluncae*	MnP, Lac, AP, CA
36	BEOFB0930000	M3	*Sarocladium spinificis*	Lac, HA, CA
37	BEOFB0260400	M3	*Talaromyces xishaensis*	-
38	BEOFB0120700	M5	*Trichoderma zeloharzianum*	-

AP—acid production; PP—pigment production; CA—cellulolytic activity; HA—hemicellulolytic activity; ligninolytic activity (Lac—laccase; LiP—lignin peroxidase; MnP—Mn-oxidizing peroxidases).

## Data Availability

Data are contained within the article.

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
