# Peer review of "Characterization of Culturable Mycobiome of Newly Excavated Ancient Wooden Vessels from the Archeological Site of Viminacium, Serbia"

_jof, 2024, doi:10.3390/jof10050343_

Round 1

Reviewer 1 Report

The manuscript focuses on culturable fungi affecting two wooden vessels excavated from river sediments. In detail, the culturable fungal diversity and their relative detrimental potential was assessed using plate assays, environmental conditions were also recorded for conservative purposes. The biodeterioration study case here reported doesn’t fit with the chosen special Issue Fungal Biodegradation: Strategies, Current Understanding, and Future Prospects: 2nd Edition because focused on biodegradation and bioremediation, a significantly different topic.

Anyway, even if the item could interest JoF readership, several concerns should be addressed before considering the manuscript for publication. In general, the organization of the manuscript needs to be improved. Additional background information is necessary to enhance comprehension of the study's framework, and further improvements must be made to the materials and methods section. And oversights amended.

As declared by the authors, before excavation, no oxygen was available, so the recorded damage cannot be justified by fungi that are aerobes. In this light, the introduction should give more details about the risks for objects undergoing sudden changes in environmental conditions. In the reported case, the deteriorative community switched from being composed of anaerobic or microaerophilic to aerobes. Otherwise, your description turns in the blink of an eye from microaerophilic conditions (87-91) to the 'deterioration capabilities of fungi' (92). Moreover, to give strength to your results, there is also a need to know the steps the objects underwent until the sampling moment (conservation treatments were mentioned in Fig. 1 caption), as well as to acquire more information about the ambient conditions at the Mammoth park (e.g., indoor, ambient dimensions, connection with the open air). Substantial improvements need to be made to figures, tables, and captions. Full accessibility to the obtained sequences as well as those used for the NJ tree must be provided (a table reporting the sequences used, species names, reference strains collection nr, refernces, etc,). In this regard, there is a methodological contradiction regarding the tree construction. Indeed, the beta-tubulin was used because it is more decisive in species-level determination (e.g. Penicillium and Aspergillus), but at the same time, figure 5 has an ITS tree.

English needs attention. Please, check grammar, words use, and fluency.

Note for the authors and Editors. In a co-authorship paper, the three individuals contributing the most to a research project are typically listed as the first, last, and second authors. The corresponding author (CA) is generally attributed to one of these scientists. Since the CA is a leading role recognized and considered in Academic evaluations, has been recorded a new trend in which “CA is used in some cases to imply, without clear justification, that author’s “leadership” of the research in analyses of multi-authored papers” (https://www.timeshighereducation.com/campus/evolving-meaning-corresponding-authorship-research - sponsored by ELSEVIER). In this light, the CA given to the author occupying the seventh position out of nine sounds inappropriate. The authors should reconsider either the authorship order or the attribution of the leading role to the CA.

Below are reported some additional comments

Title- Please add “the“ and “of”; … from the Archaelogical site of Viminacium

The abstract should be reworked. It starts with the aim missing the background information. Results and investigation methods should be also reworked

Keywords to be effective should be contextualized avoiding words already indexed from the title. Moreover, keywords are single words very rarely. In this light, Viminacium is not a good choice because already indexed. Instead of Shipwreck (too general), it could be better “buried shipwreck” or “archaeological shipwreck”

Rework sentence at lines12-15, is confusing. I also suggest removing the species names to improve reading fluency. Based on Tedersoo et al., 2018, please specify if you are either referring to Mucoromycota sensu lato ((https://link.springer.com/article/10.1007/s13225-018-0401-0) or Mucoromycota sensu strictu.

30. change “next to the stone” to alongside stone

33. organic chemical nature

32-35. Improve, it is too long

50. what about extractives?

59. the sentence sounds a bit redundant since if we are talking about microorganisms it is obvious that are biotic factors. Please simplify the sentence e.g. Wood degrading microorganisms play a leading role in wood deterioration. Moreover, it would be useful for readers if you describe the different niches of wooden objects (e.g. buried, waterlogged, aboveground) as it could be easier to evaluate the contribution of each microorganism group to deterioration based on water and oxygen availability.

74. please explain what’s holocellulose. You previously introduced cellulose, lignin and hemicellulose, this could be confusing for a student. It could be also useful explain what a monoxyl is.

89. preserved for so many years. Please change to centuries

102. remove immense, it is enough to tell that it is waiting for inclusion in the Unesco sites list.

105. Move figure 1 from results to materials and methods. The figure should be also implemented showing the geographical position of Serbia and the study site.

Consistency it is required in reporting suppliers (brand, city, country) and capital letters. Testo is a brand, the reader should be aware of it.

131. “to confirm identification” should not start a new section.

145 check spelling, monoxile

147-149, correct capital letters and in the following pages. For culture media the use of capitals is allowed because for names because it works as a title and help in memorize the relative acronyms. For the recipe components it is not because are included in the text and not in a table. Once the acronym is reported, it should be used. In this light, being carboxymethyl cellulose reported as CMC (line 168), the extended name should be avoided (172,180). Above all, since recipe reading within the text could be difficult, I suggest improving readability preparing a table. The supplier should be reported at least for the main components (i.e., Congo red, ABTS, Phenol red, Azure II etc)

184. Two different media were used to assess the ability of the isolated to degrade lignin. The media used are instead three since Mn-peroxidase contributes to lignin degradation.

193. Phenol red in a positive result turns to yellow. It does not decolorize

In situ observation are not a result, since reflect the preliminary observations before sampling- move to materials and methods. It could be useful if the sampling points are displayed in the pictures.

Figure 2. should be purged of the blank spaces and readability improved (DPI is very low). Being each figure or table a standalone material the caption should report all the elements necessary for easy understanding. the caption should be improved. Check the symbol for Celsius degrees, relative humidity is RH%. The meaning of numbers reported in X- axis is missing.

Consistently to materials and methods the identification comes before detrimental potential Figure 3 should be moved before the table. Anyway, the concerns about the use of ITS for this tree have been already raised. Each tree should be accompanied by a table listing all the strain you used for comparison (even as supplementary material). About plate test results, are you sure that there is no typos in reporting null metabolic activities for Aspergillus fumigatus? Why you did not include this genus in discussion? (lines 373-375)

Authors names are required when species are mentioned in the main text for the first time. Species authors should be included also in the tables.

Figure 4. should be reduced and improved. There is no need to report 20 plates relative to four tests only. Moreover, it is not allowed to put text over figures. The correspondence between species and plate test depicted should be reported in extent (no numbers) in the caption.

The discussion needs to be more concise and provide new insights; much of the information reported here could be useful as background material. Figures are not allowed. Beyond the need of disclosing the numbers meaning (i.e., the species) being a standalone material. This figure is a visual representation of the data already reported in Table 1. So, the authors should choose which one should be included in the Results’ main text.

298. The very high moisture of wood is one of the primary prerequisites of soft-rot fungi. This is wrong.

Soft rot fungi can be found even in dry environments and are mostly known to occur where brown- and white-rot are inhibited by factors such as high moisture content, low aeration and presence of preservatives or high temperatures (Blanchette et al., 1995; Held et al., 2005; Pournou, and Bogomolova, 2009; BRIDŽIUVIENĖ & RAUDONIENĖ, 2013only to give some examples). Indeed soft rot is common in harsh environments such as polar regions and hot deserts. Xerophilic and xerotolerant fungi (Ascomycota) have been recognized as a risk for conservation even in controlled conditions because they need low water amounts to develop (Gadd et al., 2024). 

Author Response

Reviewer #1

More details should be given in the introduction about the risks for objects undergoing sudden changes in environmental conditions. See the complete report for additional details.

A substantial change to the Introduction section was made according to the reviewers' suggestions. Point by point review is given below.

The manuscript focuses on culturable fungi affecting two wooden vessels excavated from river sediments. In detail, the culturable fungal diversity and their relative detrimental potential was assessed using plate assays, environmental conditions were also recorded for conservative purposes. The biodeterioration study case here reported doesn’t fit with the chosen special Issue Fungal Biodegradation: Strategies, Current Understanding, and Future Prospects: 2nd Edition because focused on biodegradation and bioremediation, a significantly different topic.

The authors submitted this manuscript at the invitation of the editor for the special issue Fungal Biodegradation: Strategies, Current Understanding, and Future Prospects: 2nd Edition. The editor was familiar with the title and topic of the manuscript before submission. If the editor deems it appropriate, the manuscript can be published as part of some other special issue.

Anyway, even if the item could interest JoF readership, several concerns should be addressed before considering the manuscript for publication. In general, the organization of the manuscript needs to be improved. Additional background information is necessary to enhance comprehension of the study's framework, and further improvements must be made to the materials and methods section. And oversights amended.

The manuscript was significantly improved based on the comments from both reviewers.

As declared by the authors, before excavation, no oxygen was available, so the recorded damage cannot be justified by fungi that are aerobes. In this light, the introduction should give more details about the risks for objects undergoing sudden changes in environmental conditions. In the reported case, the deteriorative community switched from being composed of anaerobic or microaerophilic to aerobes. Otherwise, your description turns in the blink of an eye from microaerophilic conditions (87-91) to the 'deterioration capabilities of fungi' (92).

More details about the risks for archeological woods that switch conditions from anoxic to aerobic are introduced in the introduction section.

Moreover, to give strength to your results, there is also a need to know the steps the objects underwent until the sampling moment (conservation treatments were mentioned in Fig. 1 caption), as well as to acquire more information about the ambient conditions at the Mammoth park (e.g., indoor, ambient dimensions, connection with the open air).

After the excavation, both objects were temporarily stored within the semi-underground exhibition space of Mammoth Park due to the COVID-19 pandemic, and no conservation work was carried out on them before sampling, as was stated in the capture of Figure 1. This is now specified in the manuscript. Details regarding the ambient conditions at Mammoth Park are now included in the manuscript (Materials and Methods section).

Substantial improvements need to be made to figures, tables, and captions.

Improvements were made to figures, tables, and captions according to the reviewer's suggestion.

Full accessibility to the obtained sequences as well as those used for the NJ tree must be provided (a table reporting the sequences used, species names, reference strains collection nr, refernces, etc,). In this regard, there is a methodological contradiction regarding the tree construction. Indeed, the beta-tubulin was used because it is more decisive in species-level determination (e.g. Penicillium and Aspergillus), but at the same time, figure 5 has an ITS tree.

A novel BenA tree was constructed and placed in the body of the manuscript as Figure 4. However, since BenA sequences are not publicly available yet, we have not provided an additional table containing the gene bank accession number of all sequences. This will be done when the sequences become publicaly available.

English needs attention. Please, check grammar, words use, and fluency.

Grammar, word use, and fluency of the English language in the manuscript were improved.

Note for the authors and Editors. In a co-authorship paper, the three individuals contributing the most to a research project are typically listed as the first, last, and second authors. The corresponding author (CA) is generally attributed to one of these scientists. Since the CA is a leading role recognized and considered in Academic evaluations, has been recorded a new trend in which “CA is used in some cases to imply, without clear justification, that author’s “leadership” of the research in analyses of multi-authored papers” (https://www.timeshighereducation.com/campus/evolving-meaning-corresponding-authorship-research - sponsored by ELSEVIER). In this light, the CA given to the author occupying the seventh position out of nine sounds inappropriate. The authors should reconsider either the authorship order or the attribution of the leading role to the CA.

The authors would like to thank the reviewer for the comment; however, all co-authors agree on their positions in the list, as well as the authors specified as the first, last, and corresponding author. With that in mind, authors reserve their discretion to determine the order and roles of authors themselves.

Title- Please add “the“ and “of”; … from the Archaelogical site of Viminacium

The title was changed according to the reviewer's suggestion.

The abstract should be reworked. It starts with the aim missing the background information. Results and investigation methods should be also reworked

The abstract was reworked according to the reviewer's suggestion.

Keywords to be effective should be contextualized avoiding words already indexed from the title. Moreover, keywords are single words very rarely. In this light, Viminacium is not a good choice because already indexed. Instead of Shipwreck (too general), it could be better “buried shipwreck” or “archaeological shipwreck”

Keywords were improved according to the reviewer's suggestion.

Rework sentence at lines12-15, is confusing. I also suggest removing the species names to improve reading fluency. Based on Tedersoo et al., 2018, please specify if you are either referring to Mucoromycota sensu lato ((https://link.springer.com/article/10.1007/s13225-018-0401-0) or Mucoromycota sensu strictu.

We have removed the species names from the sentence. We have also changed the name of Mucoromycota to Mucoromycota sensu stricto.

  1. change “next to the stone” to alongside stone

“Next to stone” was changed to “alongside stone” according to the reviewer's suggestion.

  1. organic chemical nature

“Chemical nature” was changed to “organic chemical nature” according to the reviewer's suggestion.

32-35. Improve, it is too long

The sentence was rewritten and improved to make it more clearer.

  1. what about extractives?

This correction regarding extractives in cell walls is inserted in the text.

  1. the sentence sounds a bit redundant since if we are talking about microorganisms it is obvious that are biotic factors. Please simplify the sentence e.g. Wood degrading microorganisms Wood degrading microorganisms play a leading role in wood deterioration. Moreover, it would be useful for readers if you describe the different niches of wooden objects (e.g. buried, waterlogged, aboveground) as it could be easier to evaluate the contribution of each microorganism group to deterioration based on water and oxygen availability.

The suggestion about biotic factors is accepted, and the sentence is rephrased accordingly.

  1. please explain what’s holocellulose. You previously introduced cellulose, lignin and hemicellulose, this could be confusing for a student. It could be also useful explain what a monoxyl is.

Instead of the term “holocellulose” we used “cellulose and hemicellulose” in the text to avoid confusion among readers. Additionally, it is now explained what the monoxyl is.

  1. preserved for so many years. Please change to centuries

“So many years” was changed to “centuries” according to the reviewer's suggestion.

  1. remove immense, it is enough to tell that it is waiting for inclusion in the Unesco sites list.

“Immense” was removed according to the reviewer's suggestion.

  1. Move figure 1 from results to materials and methods. The figure should be also implemented showing the geographical position of Serbia and the study site.

Figure 1 was moved to the Materials and Methods section and modified according to the reviewer's suggestion.

Consistency it is required in reporting suppliers (brand, city, country) and capital letters. Testo is a brand, the reader should be aware of it.

According to the suggestion of the reviewer all of the necessary changes were made in this part of the manuscript.

  1. “to confirm identification” should not start a new section.

The sentence was rewritten according to the reviewer's suggestion.

145 check spelling, monoxile

“Monoxile” was corrected to “monoxyle”.

147-149, correct capital letters and in the following pages. For culture media the use of capitals is allowed because for names because it works as a title and help in memorize the relative acronyms. For the recipe components it is not because are included in the text and not in a table. Once the acronym is reported, it should be used. In this light, being carboxymethyl cellulose reported as CMC (line 168), the extended name should be avoided (172,180). Above all, since recipe reading within the text could be difficult, I suggest improving readability preparing a table. The supplier should be reported at least for the main components (i.e., Congo red, ABTS, Phenol red, Azure II etc)

Capital letters and the use of abbreviations are now corrected accordingly. Furthermore, names of suppliers for nonstandard chemicals are now included in the text. In authors opinion, an additional table would unnecessarily burden the manuscript so we decided to keep recipes in the main text.

  1. Two different media were used to assess the ability of the isolated to degrade lignin. The media used are instead three since Mn-peroxidase contributes to lignin degradation.

This misconception has been corrected according to the suggestion of the reviewer.

  1. Phenol red in a positive result turns to yellow. It does not decolorize

“The decolorization zone” was changed to “the yellow zone” according to the reviewer's suggestion.

In situ observation are not a result, since reflect the preliminary observations before sampling- move to materials and methods. It could be useful if the sampling points are displayed in the pictures.

Per the reviewer's suggestion, the section regarding in situ observations was incorporated into the “Study Site and Sampling Points” subsection of Materials and Methods. Sampling points were not added to Figure 1 as they would not be clearly visible in the presented images.

Figure 2. should be purged of the blank spaces and readability improved (DPI is very low). Being each figure or table a standalone material the caption should report all the elements necessary for easy understanding. the caption should be improved. Check the symbol for Celsius degrees, relative humidity is RH%. The meaning of numbers reported in X- axis is missing.

Blank space was purged from Figure 2 and DPI was set to 300 according to the Instructions for the authors. The caption for Figure 2 was supplemented with additional information. The symbol for the Celsius degree is correct and the present underline is the style of Palatino Linotype font which is the requirement of the journal. The abbreviation for relative humidity was corrected to RH%. Numbers reported on the X-axis represent dates within the measuring period (dd.mm.yyyy).

Consistently to materials and methods the identification comes before detrimental potential Figure 3 should be moved before the table. Anyway, the concerns about the use of ITS for this tree have been already raised. Each tree should be accompanied by a table listing all the strain you used for comparison (even as supplementary material).

As addressed in the comment above, the Supplemaentary table will be provided once all Gene Bank Accession numbers become publicly available.

About plate test results, are you sure that there is no typos in reporting null metabolic activities for Aspergillus fumigatus? Why you did not include this genus in discussion? (lines 373-375)

In 7 performed plate assays Aspergillus fumigatus lacked any activity; hence, it was not included in the Discussion, as the main focus was on fungi with positive results obtained in the experiment.

Authors names are required when species are mentioned in the main text for the first time. Species authors should be included also in the tables.

Since the research presented in the manuscript is not taxonomic, the authors feel that there is no need to introduce species authorship and additionally burden the main text.

Figure 4. should be reduced and improved. There is no need to report 20 plates relative to four tests only. Moreover, it is not allowed to put text over figures. The correspondence between species and plate test depicted should be reported in extent (no numbers) in the caption.

Figure 4 and the caption were modified according to the reviewer's suggestion.

The discussion needs to be more concise and provide new insights; much of the information reported here could be useful as background material. Figures are not allowed. Beyond the need of disclosing the numbers meaning (i.e., the species) being a standalone material. This figure is a visual representation of the data already reported in Table 1. So, the authors should choose which one should be included in the Results’ main text.

The discussion was reworked according to the recommendations from both reviewers, and Figure 5 was removed from the manuscript.

  1. The very high moisture of wood is one of the primary prerequisites of soft-rot fungi. This is wrong. Soft rot fungi can be found even in dry environments and are mostly known to occur where brown- and white-rot are inhibited by factors such as high moisture content, low aeration and presence of preservatives or high temperatures (Blanchette et al., 1995; Held et al., 2005; Pournou, and Bogomolova, 2009; BRIDŽIUVIENĖ & RAUDONIENĖ, 2013only to give some examples). Indeed soft rot is common in harsh environments such as polar regions and hot deserts. Xerophilic and xerotolerant fungi (Ascomycota) have been recognized as a risk for conservation even in controlled conditions because they need low water amounts to develop (Gadd et al., 2024).

This part of the discussion was reworked precisely according to the recommendation from the reviewer.

Reviewer 2 Report

The Authors of the work presented publications on the microbiota of fungi isolated from archaeological objects. In their research, the authors of the work identified fungi and indicated their biochemical features related to the ability to produce enzymes, acids and pigments. The tests for bochemical characteristics were carried out based on specific tests on agar media.

The Authors should know and emphasize it in the discussion that the production of metabolites (dyes, acids, enzymes and others) depends on the substrate on which mold fungi grow. Identifying dyes on an agar medium is important, but it means that the fungus will also produce dye on wood samples.

In the description of the conclusions, the authors of the study raised a very important issue: "..with results used as the indicator in the selection of the most appropriate conservation treatments" but it should be remembered that the problems of archaeological wood conservation are not related to the microorganisms themselves, but primarily to the problems durability and stability of such wood. Therefore, preservation of such wood against biodegradation is different than that of regular wood.

From the point of view of the assessed wood samples, these tests are important, but whether they have a universal impact on other objects is unlikely.

After minor changes in the discussion and conclusions, I recommend the work for publication.

        ​Do the numbers marked in Figure 4 represent the fungal strains numbered in Table 1?          

Author Response

Reviewer #2

The conclusions are described clearly, but due to the fact that the authors of the study indicate that the identification of pathogens on wood samples will allow the use of appropriate methods and means of protection against corrosion, this statement is far from exaggerated. The problems of preserving archaeological wood are largely related to its structure and state of preservation. The place where the wood was stored also influences the impregnation process, which may be related to the presence of certain micro and macro elements in the wood, which may determine maintenance problems.

The authors agree with the statement made by the reviewer; however, more precisely the results were meant to be used in selecting the most appropriate conservation treatments to alleviate post-excavation fungal infestation, not conserving the wood in general as it depends on numerous other factors. This is now made clear in the Conclusions section of the manuscript.

The Authors of the work presented publications on the microbiota of fungi isolated from archaeological objects. In their research, the authors of the work identified fungi and indicated their biochemical features related to the ability to produce enzymes, acids and pigments. The tests for bochemical characteristics were carried out based on specific tests on agar media. The Authors should know and emphasize it in the discussion that the production of metabolites (dyes, acids, enzymes and others) depends on the substrate on which mold fungi grow. Identifying dyes on an agar medium is important, but it means that the fungus will also produce dye on wood samples.

The authors agree with the statement made by the reviewer and have made appropriate changes to in Conclusion section of the manuscript to reflect on that.

In the description of the conclusions, the authors of the study raised a very important issue: "..with results used as the indicator in the selection of the most appropriate conservation treatments" but it should be remembered that the problems of archaeological wood conservation are not related to the microorganisms themselves, but primarily to the problems durability and stability of such wood. Therefore, preservation of such wood against biodegradation is different than that of regular wood.

The authors agree with the statement made by the reviewer and have made appropriate changes to in Conclusion section of the manuscript to reflect on that.

From the point of view of the assessed wood samples, these tests are important, but whether they have a universal impact on other objects is unlikely.

As previously stated, the authors agree with this statement made by the reviewer and have made appropriate changes to in Conclusion section of the manuscript to reflect on that.

After minor changes in the discussion and conclusions, I recommend the work for publication.

The authors would like to thank reviewer #2 for the recommendation for the publication of the manuscript!
